# Neural signatures of bullying experience and social rejection in teenagers

**Markus Kiefer**[1]*, **Eun-Jin Sim**[1], **Sabrina Heil**[1], **Rebecca Brown**[2], **Bärbel Herrnberger**[1], **Manfred Spitzer**[1,3], **Georg Grön**[1]

**1** Department of Psychiatry and Psychotherapy III, Ulm University, Ulm, Germany, **2** Department of Child and Adolescent Psychiatry and Psychotherapy, Ulm University, Ulm, Germany, **3** Transfer Center for Neuroscience and Learning, Ulm University, Ulm, Germany

☯ These authors contributed equally to this work.
* Markus.Kiefer@uni-ulm.de

## Abstract

Relational bullying in schools is one of the most frequent forms of violence and can have severe negative health impact, e.g. depression. Social exclusion is the most prominent form of relational bullying that can be operationalized experimentally. The present study used MR-based perfusion imaging (pCASL) to investigate the neural signatures of social exclusion and its relationship with individually different extent of previous bullying experience. Twenty-four teenagers reporting bullying experience at different extent were scanned during a virtual ball-tossing (Cyberball game). Our findings showed that social exclusion (relative to social inclusion) activated frontal brain areas: sub- and perigenual anterior cingulate cortex (sg/pgACC), left inferior frontal cortex (IFG), and dorsolateral prefrontal cortex. Positive relationship between exclusion-specific signal increase and individually different extents of prior bullying experience was for the first time observed in left IFG and sgACC. This suggests that more frequent prior experience has conditioned greater mentalizing and/or rumination, in order to cope with the situation. While this interpretation remains speculative, the present data show that the experience of being bullied partly sensitizes the neural substrate relevant for the processing of social exclusion.

## Introduction

Bullying is one of the most common forms of violence in schools [1]. Olweus, who was first investigating this issue systematically [2,3] and developed a bullying/victim questionnaire [Bullying Questionnaire, QBQ, 4], defined bullying as a specific form of aggression [5]. It is characterized by an imbalance of power, whereby a more powerful individual repeatedly and intentionally causes harm to a weaker individual [6]. Bullying at schools is a worldwide observation. For instance, about 18.7% of students across OECD countries reported that any type of bullying act occurred at least a few times a month in 2015 [7], and around 23% reported the same incidence in 2018 [1]. A further increase in school-aged children's exposure to bullying is therefore to be expected.

**Data Availability Statement:** Data cannot be shared publicly because of the European General Data Protection Regulation (GDPR), which does not allow transfer of biological material including brain imaging data without explicit consent of the

participants. However, data are available from the corresponding author (contact via markus.kiefer@uni-ulm.de) or from the Data management assistant at the Section for Cognitive Electrophysiology within the Department of Psychiatry III in the University Ulm (contact via silvia.zischler@uni-ulm.de) for researchers who meet the criteria for access to confidential data by the Ethics Committee (contact via ethic-kommission@uni-ulm.de).

**Funding:** The authors received no specific funding for this work.

**Competing interests:** The authors have declared that no competing interests exist.

Bullying has been treated as a social phenomenon at school days for a long time [8,9]. However, accumulating evidence in the last years especially highlighted the negative consequences on mental health of the bullied victim rather than treating bullying as a mere social phenomenon within and across peers [10]. For example, recent studies have shown that bullying has severe negative impact on the victim during school life and is related to high social anxiety [11,12], low self-esteem [13], loneliness and sadness [14,15], and depressive symptoms [7,16–19]. Klomek et al. [20] particularly suggested that bullying is one of various factors relevant for maintenance of depression in children and adolescents. Other studies investigating the relationship between bullying and depression, showed more depressive symptoms and psychological distress in victims than in non-victims [18,21–27]. Schwartz et al. [28] also provided insight into the role of peer victimization in the prediction of depressive symptoms. Therefore, emotional instability and behavioral problems suffered by victims may continue into adulthood, leading to long-term negative consequences [29]. Despite these insights, prevention programs against bullying and its consequences showed only moderate success so far suggesting room for further improvements [30,31]. This imposes the necessity to better understand the cognitive and neural mechanisms associated with bullying experience.

One of the main offending features of relational bullying is social exclusion from the peer group. Neuroscientific research on social exclusion might therefore provide insights into the neural and cognitive mechanisms that lead to negative emotional consequences in bullied victims. The well-established virtual ball-tossing game [Cyberball game; 32] is an animated experimental paradigm that simulates experience of social exclusion [33]. This game has been used in several neuroimaging studies to elicit the experience of social exclusion among healthy adolescents [34–38], specifically with non-suicidal self-injury [39], and healthy adults [40–42]. In that game, the participant first plays the ball with two other virtual players via a computer (the social inclusion condition). After some initial ball tosses involving the participant, the virtual players suddenly exclude the participant during the remainder of the game (the social exclusion condition). The first study investigating the Cyberball game using functional magnetic resonance imaging [41] revealed stronger neural activation of the anterior cingulate cortex (ACC) during social exclusion than social inclusion in healthy adults. Moreover, the magnitude of neural activation of the ACC mediated the correlation between right ventral prefrontal cortex (VPFC) and self-reported distress. Other studies with the same paradigm also highlighted the commensurability of experiences of social and physical pain [43–45]. A meta-analysis [46] comprising 46 articles from 2003 to 2013 investigated the role of ACC subdivisions for the processing of social rejection. The authors found that self-reported distress was associated with activation levels of the subgenual anterior cingulate cortex (sgACC) and pregenual ACC (pgACC) in painful social tasks, such as the exclusion situation of the Cyberball game. Involvement of different ACC subdivisions during the social exclusion condition was also shown to correlate with personality traits and dispositions (e.g. Masten et al., 2013), indicating that individually different experiences may modulate neural social exclusion signaling. In this context, Masten et al. [47] found that healthy adolescents, who reported high sensitivity to being socially rejected before the MR scanning, showed stronger neural activation consistent with experiencing social distress when observing a peer being excluded. This finding indicated that the rejection sensitivity as a disposition might induce neural activation of precuneus, ventrolateral PFC (vlPFC), and notably the sgACC, which is involved in painful situations, even though this situation was displayed just virtual.

While there was a clear effect of bullying sensitivity as a modulating disposition, participants in this study had not been asked whether they had previously experienced bullying in real school life. For that reason, the modulating effect of prior real life bullying experiences on the neural processing of being excluded remains to be investigated.

Against this background, the purpose of the present study was to investigate the relationship between differential brain activation due to social exclusion (vs. social inclusion) and its relation to individually different experiences of bullying in adolescence. Brain activation during social exclusion was assessed using MR-based perfusion imaging using a 3D pseudo-continuous arterial spin labeling (pCASL) technique [48], which measures regional cerebral blood flow (CBF) during the different conditions of the Cyberball game [32]. Moreover, previous exposure to bullying was measured by the Bullying/Victimization Questionnaire for children [BVF-K;49]. We also measured the extent of depressive symptoms to investigate the possibility that adolescent's exposure to bullying may be related to depression. We hypothesized that prior bullying experience among teenagers would be related to brain activity in response to social exclusion compared to social inclusion. Given the exploratory character of our study with regard to the neural mechanisms involved in bullying experience and social exclusion in teenagers, we could not suggest a directional hypothesis for the association of bullying experience with neural responses during the Cyberball task. As the positive relation between bullying and depression is well established [20], we expected that exposure to bullying would be positively related to depressive symptoms in teenagers.

## Materials and methods

### Participants

Participants were 29 schoolchildren aged between 12 and 15 years (M = 13.5), who visited middle school or high school at that time. All participants were recruited through fliers distributed in the city of Ulm, Germany, between April 2017 and February 2018. Exclusion criteria were a medical, neurological or psychiatric disorder. Participants originally recruited for the study, but with a history of neurological disorder, brain injuries or neuroanatomical abnormalities (N = 3) were excluded from data analysis. Furthermore, two participants were excluded due to excessive head movements greater than 1.5 mm during MR imaging, resulting in a final sample of 24 participants. Of the remaining 24 participants (M/F = 10/14), four were left handed. All of them had normal or corrected-to-normal vision. Due to the novelty to engage MR-based perfusion imaging of the Cyberball-task in combination with schoolchildren as participants, who were also likely to have experienced some kind of bullying, we were not able to obtain effect sizes before running our study. For that reason, we could not estimate the appropriate sample size to yield sufficient statistical power in an apriori power analysis. Consequently, the rationale for setting the sample size was driven by a report of the median sample size in fMRI studies of around 29 participants in a previous review article [50].

Participants voluntarily took part, after providing written informed consent of their own and their parents or guardian. At least one of the parents or legal guardians was present during the entire study. Participants were compensated by 30 Euros. Both the regional council of state education authority in Tübingen (Baden-Württemberg, Germany), and the ethics committee of Ulm University, Germany, approved the study (Ref.: 332/2015). All procedures were in accordance with the ethical standards of the institutional and/or national research committee and in line with the 2013 Helsinki declaration [51] and its later amendments or comparable ethical standards.

### Questionnaires

Prior to perfusion MR scanning, the participants completed a depression questionnaire for children [DIKJ; 52] and the BVF-K [49]. DIKJ, which has been developed based on the Kovacs Children's Depression Inventory [CDI; 53,54] in a German version, is a reliable and valid measuring instrument [Split-Half-Reliability: r = 0.82, Cronbach's α = 0.84; 55,56] to assess current

depressive symptoms in children (10–16 years). DIKJ consists of 29 items and each item was rated on a three-category Likert scale from 0 (not at all) to 2 (very much). The sum score reflects severity of depressive symptoms. Sum score $\geq$ 18 of DIKJ indicates clinically relevant depression. BVF-K [49] measures previous exposure to bullying and victimization. BVF-K is a validated instrument (internal consistency Cronbach's $\alpha$ = 0.77 to 0.90 [cf. 57] based on self-reports for school-aged children. Each scale has four subscales (direct/indirect relational victimization and direct/indirect behavioral aggression) which consist of eight items. Each item describes a different behavior, and the participant was asked to determine from a three-category Likert scale ("none", "sometimes", and "often") the frequency of that behavior over the past month. We calculated sum scores for the bully and victim scales (range: 0–16 for each). To check the ostracism manipulation, we used the Need-Threat-Scale [NTS; 58] after translating the original English version into German [see also 39]. The full need-threat scale was adapted from previous studies [59,60]. NTS is a well-established questionnaire and is frequently used for measuring distress after social exclusion elicited by the Cyberball task. It consists of 20 items assessing the need for belonging, self-esteem, meaningful existence, and control with five items each. Internal consistency of the need- threat scale proved to be good [Cronbach's $\alpha$ = 0.78, 61–63]. For the present study, we used the belonging scale of the NTS after the first run of each Cyberball game condition, in order to measure the magnitude of participants' current experience, for example "I felt I belonged to the group". Each item was rated on a scale from 1 (not at all) to 5 (extremely). We calculated the sum of the ratings for each condition specific repetition of this scale. Higher sum scores thus indicated more satisfaction of need (i.e., feeling more included). Therefore, the belonging scale of NST was evaluated three times, after the first run of each Cyberball game condition inside the scanner room, whereas the other subscales of the NTS were completed once after the last run of the Cyberball game outside the scanner.

## Experimental task

After completing the questionnaires (DIKJ, BVF-K), participants performed a Cyberball task as the experimental task during MR imaging. The Cyberball task [32] consists of three conditions, which are typically run in a fixed order: Participants are instructed to only watch the ball-tossing between three virtual players on the screen (condition 1: passive viewing). Then, participants are instructed to toss the ball to the two virtual players on the upper-right and upper-left part of the screen (condition 2: social inclusion). Finally, participants are instructed to play the game exactly like the second run. That last condition starts with 10 throws of 30% randomized ball-possession. Thereafter, participants are excluded from the game and only the virtual players toss the ball to each other (condition 3: social exclusion). This paradigm was used in an earlier fMRI study by our group [37]. For the present perfusion study, we used it with the following modifications: (1) In order to obtain sufficient quality of perfusion signals, participants played the Cyberball game three times, i.e. there were three runs of each condition. (2) For each participant the first run was always the condition of passive viewing, for measuring baseline perfusion. The other eight runs were presented in pseudo-randomized order, but with the restriction that the same condition was never presented twice in direct sequence. Participants were explicitly informed about the purpose of the Cyberball game after the end of perfusion measurement. The Cyberball game was programmed using the Presentation® software package (Version 14.8, Neurobehavioral Systems, Inc., Berkeley, CA).

## MRI data acquisition

Magnetic resonance imaging (MRI) data were acquired with a 3-Tesla MAGNETOM Prisma scanner in combination with a standard 64-channels head/neck coil (Siemens, Erlangen,

Germany) at the Department of Psychiatry of Ulm University. Participants were placed in the MR scanner with their head padded to minimize movement artefacts during data acquisition. Participants were in permanent contact with the experimenter during acquisition and could interrupt it at any time. Pictures were projected onto a 32" LCD screen (NordicNeuroLab AS, Bergen, Norway) located behind the scanner, which participants could view through a mirror mounted on the MRI head coil. Regional cerebral blood flow (CBF) served as a marker for energy-intensive neuronal activity and was measured using MRI-based pseudo-continuous arterial spin-labeling (pCASL) technique. The applied 3D gradient-echo spin-echo imaging sequence was developed by the German Center for Neurodegenerative Diseases (DZNE), Bonn, Germany, and used with parameters based on recommendations by Alsop et al. [64] for MRI-based perfusion imaging without contrast agents: TR/TE: 3900/20 ms, matrix 64 x 64, field-of-view (FOV) 224 mm, 40 slices, slice thickness: 2.5 mm with a gap of 1 mm, transverse slice positioning along the AC-PC line, ascending slice acquisition, flip angle 90˚, PAT factor 2 (GRAPPA mode), bandwidth 2298 Hz/pixel, marking duration (bolus length) 2400 ms, transit time (post-labeling delay) 1000 ms, pre-saturation and background suppression (BS) switched on, BS method 2 pulses. The calibration measurement took 28 seconds. Each perfusion sequence lasted 85 seconds and acquired five label and five control images for later calculation of regional cerebral blood volume.

Two structural MRI measurements were carried out after the Cyberball task. T1-weighted anatomical recordings were obtained by 3D magnetization-prepared rapid gradient echo sequence (MP-RAGE) in sagittal direction with TR/TE 2300/2.32 ms, inversion time 900 ms, flip angle 8˚, matrix 256 x 256, FOV 240 mm, 192 slices, slice thickness 0.9 mm, distance factor 50%, and layer oversampling 16.7%. The measurement time was 5 minutes 21 seconds. T2-weighted anatomical images were taken transversely using a turbo spin echo sequence with TR/TE 6540/92 ms, flip angle 150˚, matrix 320 x 320, FOV 210 mm, 30 slices, slice thickness 3.5 mm, and distance factor 15%. The measurement time was 1.59 minutes.

## Data analyses

### Questionnaires

The data of the DIKJ, BVF-K, and NTS were analyzed using SPSS (IBM SPSS Statistics, version21). Sum scores were generated per person and per questionnaire. Differences before and after each Cyberball condition were evaluated with paired two-tailed t-tests using the responses of the NTS belonging scale. All statistical tests were carried out at the significance level of $\alpha$ = .05.

### MRI data

Preprocessing of the imaging data and statistical analyses were conducted with the Statistical Parametric Mapping software package (SPM12 r6685, Wellcome Department of Cognitive Neurology, London, United Kingdom,) and an in-house toolbox hosted under SPM and based upon Perf_resconstruct_V02 SPM add-on software by H.Y. Rao and J.J. Wang, from the Department of Radiology and Center for Functional Neuroimaging at University of Pennsylvania, Philadelphia, PA, USA (https://cfn.upenn.edu/perfusion/software.htm), running on MATLAB (version: 7.9.0.529; R2009b, MathWorks Inc., Natick, Massachusetts, USA). The first step of preprocessing aimed at motion correction of each experimental time series in the Cyberball task. Motion correction is a gray value-based operation of minimizing the squared distance [65] between two volumes that follow one another in time [see also 66]. In a next step, an average volume was calculated for each individual motion-corrected perfusion series, coregistered to the average volume of the first perfusion series, and respective parameters of

coregistraton applied to volumes of a series. In the end, all perfusion volumes of a participant were in the space of the average volume of the first perfusion series. Next, in each series, regional cerebral blood flow (CBF) in units of ml/100g/minute was calculated from the difference of control minus labeled volumes following Alsop et al. [64] where

$$CBF = \frac{6000 \cdot \lambda \cdot (I_C - I_L) \cdot e^{\frac{d}{T_1}}}{2 \cdot \alpha \cdot T_1 \cdot I_{PD} \cdot \left(1 - e^{-\frac{\tau}{T_1}}\right)}$$

with brain-blood partition coefficient $\lambda = 0.9$ ml/g, longitudinal relaxation time of blood $T_1 = 1.65$ s at B0 = 3 Tesla, post-labeling delay $d = 2400$ ms, labeling $\alpha = 0.85$, and label duration $\tau = 1000$ ms. IC, IL, and IPD are the signal intensities of control, labeled and proton-density weighted images, respectively.

The CBF volumes were then normalized into the Montreal Neurological Institute (MNI) stereotactic standard space, for inter-individual comparability and analysis over group. Before normalization, the 3D-T1 volume was co-registered to the average volume of the first perfusion time series. Afterwards the normalization parameters were applied to the co-registered 3D-T1 volume and all CBF volumes. Finally, the normalized volumes were spatially smoothed using a Gaussian 3D filter with FWHM of 8 mm. For individual first level analyses, these volumes were entered into the data matrix of a general linear model. Its coefficient estimates constituted an average CBF volume for each combination of condition (passive, inclusion, exclusion) and repetition (first, second, third), after grand mean scaling to the standard unit of 50 ml/100 g/minute. These average CBF volumes were the input for second level group analysis in which participants were modeled as a random effect. Two t-contrasts were then determined within this model: inclusion vs. exclusion and exclusion vs. passive viewing, whereby the change in CBF was averaged over repetitions. A threshold of p = 0.05, family-wise error rate (FWE)-corrected with cluster size of k > 25 voxels was used for inference of significant clusters in a whole brain analysis. Images for the figures were generated using the free software package MRIcroN (https://www.nitrc.org/projects/mricron/).

Associations between the neural activation in response to social exclusion and individual scores from the different scales of interest were computed using two separate linear regression analyses: (1) rating scores of the subjective feeling of social exclusion (of being rejected), and (2) sum of individual victim scores. Individual average CBF volumes of the contrast exclusion minus inclusion were the dependent variable. The regressor of subjective feelings during social exclusion was taken from the responses of the belonging items of the NTS, and was calculated by subtracting the sum score of the five items after the exclusion condition from the sum score of the five items after the inclusion condition. Higher values therefore correspond to a higher subjective feeling of social exclusion compared to social inclusion, and represent higher subjective feeling of being rejected. The victim score per participant corresponds to the sum of the raw scores of item numbers 2 to 9 of BVF-K (victim scores of BVF-K). Linear regression analyses for these two regressors were restricted to the peak voxels of the main effect of exclusion minus inclusion (FWE: p < 0.05, k > 25) applied as an inclusive mask. Individual values from significant peak voxels (p < 0.05, uncorrected) were extracted and Pearson correlation coefficients between individual perfusion changes and the predictor of interest computed.

## Results

### Questionnaires

Participants showed higher values for the victim (M: 1.67) than for the bully scale (M: 0.67) in the BVF-K. Mean sum score of the DIKJ was 9.75 (see also Table 1). Two participants (8%)

Table 1. Demographic data and descriptive statics of questionnaires.

|  | Total sample (N = 24) | Boys (N = 10) | Girls (N = 14) |
|---|---|---|---|
| Age (range) | 13.5 (12–15) | 13.70 (12–15) | 13.29 (12–15) |
| Victim score of BVF-K (SD) | 1.67 (1.79) | 1.50 (1.18) | 1.79 (2.15) |
| Bullying score of BVF-K(SD) | 0.67 (1.01) | 1.0 (1.25) | 0.43 (0.76) |
| DIKJ (SD) | 9.75 (6.30) | 11.60 (7.15) | 8.43 (5.52) |
| NTS belonging scale before exclusion (SD) | 4.62 (0.41) | 4.56 (0.37) | 4.65 (0.5) |
| NTS belonging scale after exclusion (SD) | 1.38 (0.48) | 1.44 (0.55) | 1.34 (0.37) |

showed indication of depression (sum score $\geq$ 15 on DIKJ, for details see Stiensmeier at al., 2000). There was no significant correlation between depressive symptoms (DIKJ) and victim scores (r = -0.008, p = 0.97).

Considering participants' feelings following experience of social exclusion during the Cyberball game, the mean score of the NTS belonging scale from 1 (not at all) to 5 (extremely) was 4.62 (range: 3.4–5) before the social exclusion game and dropped to 1.38 after experience of social exclusion (range: 1–3). The participants felt greater belonging during social inclusion than exclusion (t (23) = 24.55, p < 0.0001, Cohen's d = 5.01). These scores did not differ for gender, neither after the inclusion condition (t (22) = 0.5, p > 0.1, Cohen's d = 0.24) nor after the exclusion condition (t (22) = -0.48, p > 0.1, Cohen's d = 0.24). In order to explore whether changes of feeling of belonging before and after the Cyberball game were associated with the extent of exposure to bullying, we additionally computed the Pearson correlation coefficient between the difference of the belonging scale and the victim scale, which revealed no significant result (r = -0.18, p = 0.41). After completion of the experimental sessions, we asked all participants whether they had believed playing with two partners during the gaming session. All participants reported that they believed that they had played with two virtual partners. Furthermore, all participants felt to be included or excluded by their partners in the corresponding blocks. None of the participants reported any scepticism with regard to the presence of their virtual partners.

## MRI data

**Group analysis in contrast exclusion vs. inclusion.** In the contrast of exclusion minus inclusion several significant clusters were observed (p < 0.05; FWE-corrected, k > 25). The largest cluster included subgenual ACC (sgACC) near the orbitofrontal cortex and pregenual ACC (pgACC) (see Fig 1A and 1B). The inferior frontal gyrus (IFG) including Broca's area (Fig 1C) and the margin of the left insula (AI), dorsolateral prefrontal cortex (DLPFC), and medial prefrontal cortex (MPFC) were also more strongly activated during exclusion than inclusion, as were the superior temporal gyrus and the temporal pole (see also Table 2).

**Group analysis in contrast inclusion vs. exclusion.** At the same level of significance as indicated above, the inverted contrast of inclusion minus exclusion revealed changes of neural activation in the precuneus and in the superior parietal lobe (SPL) bilaterally. Bilateral frontal eye fields (FEF) located in the superior frontal gyrus, supplementary motor area (SMA), post-central gyrus, middle temporal cortex extending to fusiform gyrus (FFG), bilateral occipital regions (V5/MT), and left cerebellum were also activated more strongly during the inclusion condition relative to exclusion.

**Association of subjective feelings of being rejected with the perfusion changes in the contrast exclusion vs. inclusion.** There were significant positive relationships (p < 0.05 at the peak voxel) of the subjective feelings of being rejected with the perfusion changes in brain regions

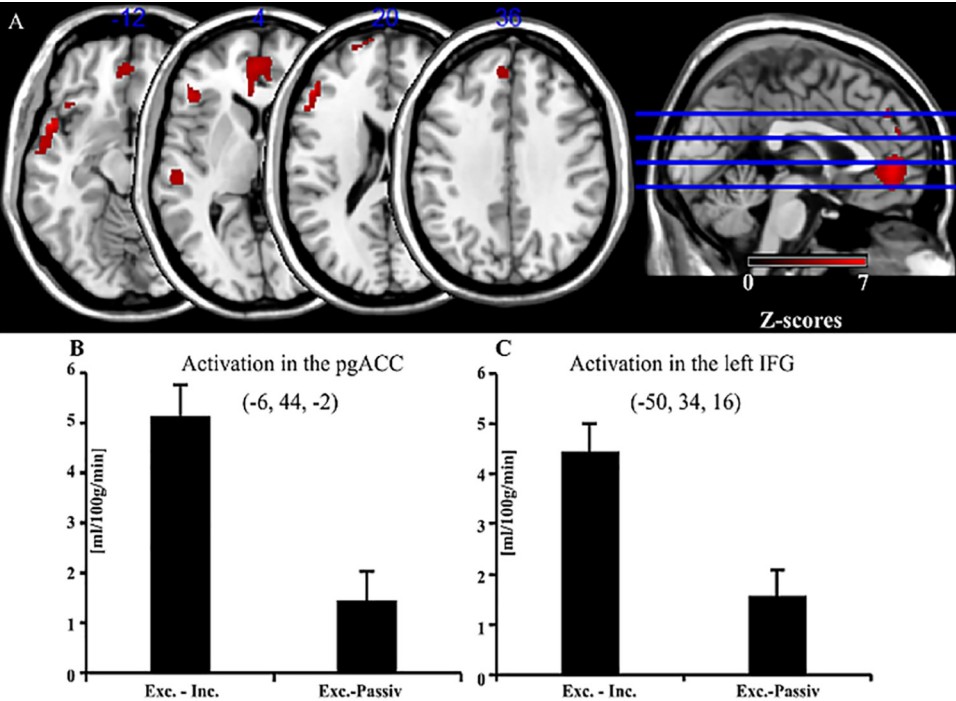

**Fig 1. Main effect: Exclusion minus inclusion in the Cyberball game.** (A) Main effect: Exclusion minus inclusion in the Cyberball game. (B-C) Bar graphs show average perfusion changes and standard errors of peak voxels in sgACC and left IFG in the contrasts of exclusion—inclusion and exclusion—passive viewing. Abbreviations: A.: pgACC, pregenual anterior cingulate cortex; IFG, inferior frontal gyrus. B-C.: Exc, Exclusion; Inc., Inclusion; Passiv, Passive viewing. While not of primary interest, differences between social exclusion and passive viewing were plotted as a reference to show that the differences between social exclusion and social inclusion were diminished since passive viewing is already different from the inclusion condition along this dimension, that is, passive viewing has more in common with the exclusion condition than with the inclusion condition.

sensitive to social exclusion (contrast exclusion minus inclusion). The more the feeling of being rejected the greater was differential neural activation in left IFG (Table 3, see also Fig 2A).

**Association of victim scores on BVF-K with the perfusion changes in contrast exclusion vs. inclusion.** Table 4 summarizes significant positive correlations ($p < 0.05$ at the peak voxel) of participants' victim scores with the perfusion change in brain regions sensitive to social exclusion (contrast of exclusion minus inclusion).

Correlation coefficients ranged between 0.48 and 0.69: Higher values of the victim scores on the BVF-K were associated with greater changes of perfusion signal in left IFG and sgACC (see Fig 3, blue). These results revealed that greater perfusion changes in the exclusion condition during the Cyberball game were related to the extent of bullying exposure. Note, that differential activation of left IFG was positively correlated with both the BVF-K victim scores ($r = 0.69$ at peak activation at [x, y, z]: [−48, 38, 6]; z = 3.70) and subjective feelings of exclusion ($r = 0.64$ at peak activation at [x, y, z]: [−50, 28, 6]; z = 3.37. see Fig 2). This brain area was therefore associated with both bullying exposure and increased subjective feelings of being rejected.

## Discussion

The present study investigated brain activation associated with bullying experience among teenagers using MR-based perfusion imaging (pCASL). Social exclusion as one core feature of

**Table 2. Main effect: Exclusion minus inclusion in the Cyberball game.**

| Anatomical Region | BA | x | y | z | z-value |
|---|---|---|---|---|---|
| sgACC/pgACC | 32 | -6 | 44 | -2 | 6.55 |
| | | 6 | 40 | 4 | 4.79 |
| Left temporal pole | 38 | -52 | 10 | -12 | 6.53 |
| | | -56 | -2 | -8 | 5.80 |
| Left IFG, pars triangularis | 44/45 | -50 | 34 | 16 | 6.51 |
| | | -48 | 30 | 4 | 5.71 |
| | | -54 | 22 | 20 | 5.54 |
| Left temporal pole | 38 | -42 | 22 | -18 | 5.52 |
| | | -30 | 26 | -18 | 4.65 |
| Right temporal pole | 38 | 42 | 20 | -24 | 5.48 |
| Left middle temporal gyrus | 22 | -54 | -24 | 2 | 5.24 |
| Left frontal superior gyrus | 9 | -6 | 46 | 34 | 5.06 |
| Left frontal superior gyrus | 32 | -2 | 52 | 24 | 4.61 |
| Left dorsolateral MFG | 10 | -12 | 66 | 18 | 5.00 |
| | | -20 | 62 | 22 | 4.75 |

Significant peak activations of exclusion minus inclusion during the Cyberball game at a statistical threshold of p < 0.05 family wise error rate (FWE)-corrected at the voxel level, at cluster extent k > 25. Listed are peak voxels with highest z-values for significant clusters and their local maxima more than 8 mm apart.

relational bullying was experimentally realized by a virtual ball-tossing game task [Cyberball game; 32] with social inclusion as the control condition. We examined whether significant exclusion-specific signal changes relate to individually different levels of prior experience with being bullied.

Participants reported greater subjective feelings of being socially excluded in the social exclusion condition, relative to social inclusion. This shows that the Cyberball game led to expected feelings of exclusion in the corresponding condition. Contrary to previous studies Klomek et al. [20], however, an association between the extent of bullying experience and severity of depressive symptoms was not observed. This might be explained by the observation that (1) all but two participants did not show clinically relevant depressive symptoms and that (2) there was a rather small extent of bullying experience among participants. The sum scores of bullying exposures as a victim (victim subscale of BVF-K) varied between 0 and 6, while the scale permits ranges between 0 and 16.

At the neural level, social inclusion relative to social exclusion yielded significant differential activation of the superior parietal lobe bilaterally, bilateral frontal eye fields, supplementary motor area, middle temporal cortex extending into fusiform gyrus, and bilateral occipital

**Table 3. Correlations between scores of subjective feeling of being rejected and perfusion changes in contrast exclusion minus inclusion.**

| Anatomical Region | BA | Z-value | x | y | z | r |
|---|---|---|---|---|---|---|
| Left IFG, pars triangularis | 45 | 3.37 | -50 | 28 | 6 | 0.64 |
| | | 3.14 | -50 | 34 | 16 | 0.61 |
| | | 2.90 | -46 | 32 | 0 | 0.57 |

BA = Brodmann area: r = Pearson correlation coefficient: The correlations with the peak voxels are significant for $p < .05$ (uncorrected).

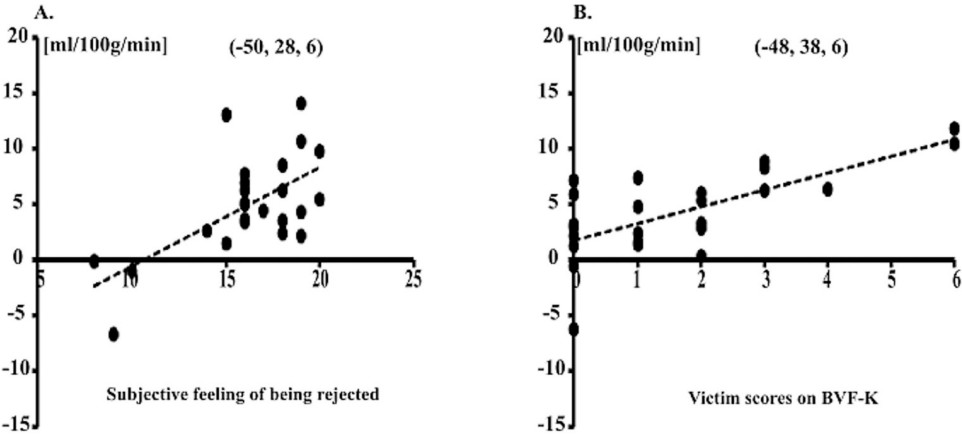

**Fig 2. Linear relationships between CBF and subjective feeling of being rejected and between CBF and victim scores.** (A) Positive linear relationship between subjective feeling of being rejected (x-axis) and CBF changes in the contrast of exclusion minus inclusion (y-axis) in left IFG at the peak voxel. Greater change of CBF was significantly associated with higher scores for the subjective feeling of being rejected. (B) Positive linear relationship between victim scores on BFV-K (x-axis) and CBF changes in the contrast of exclusion minus inclusion (y-axis) in left IFG at the peak voxel. Greater change of CBF was significantly associated with higher victim scores.

regions (V5). These regions are typically involved in various aspects of visuo-motor processing [67]. This activity pattern is in accordance with the affordance of this Cyberball game condition, in which participants were actively engaged in ball-tossing with the virtual players.

Social exclusion relative to social inclusion activated frontal—subgenual and perigenual anterior cingulate cortex, left inferior frontal gyrus, left dorsolateral frontal cortex, medial prefrontal cortex -, and temporal regions. Compared with previous results from this task in adults and adolescents [34,41,47], the present findings of involvements of sub- and perigenual anterior cingulate cortices, middle lateral prefrontal cortex and medial prefrontal cortex are in good agreement, even though a different functional imaging method has been applied to measure neural activation (perfusion vs. BOLD fMRI). Particularly, the subgenual anterior cingulate cortex has been reported to be active in studies with adolescents [34,35,37,47], where peer rejection and accompanying social distress were of primary interest. Since the very same brain region has also been involved in rumination [68], i.e. self-referential internal commenting in service of coping with usually negatively connoted issues [69], emotion regulation may be the driving factor for its activation during the social exclusion condition of the Cyberball game [see also 70]. This interpretation is further supported by the positive correlation between signal change in subgenual anterior cingulate cortex and individual prior experience of being bullied. That is, the more frequent participants had to cope with this experience in the past, the more

**Table 4. Correlations between victim scores of BVF-K and perfusion changes in contrast exclusion minus inclusion.**

| Anatomical Region | BA | Z-value | x | y | z | r |
|---|---|---|---|---|---|---|
| Left IFG. p. triangularis | 45 | 3.70 | -48 | 38 | 6 | 0.69 |
| | | 3.20 | -40 | 30 | 8 | 0.62 |
| | | 2.61 | -52 | 30 | 20 | 0.52 |
| sgACC | 11 | 2.38 | -8 | 36 | 2 | 0.48 |

BA = Brodmann area: r = Pearson correlation coefficient: The correlations with the peak voxels are significant for *p* < .05 (uncorrected).

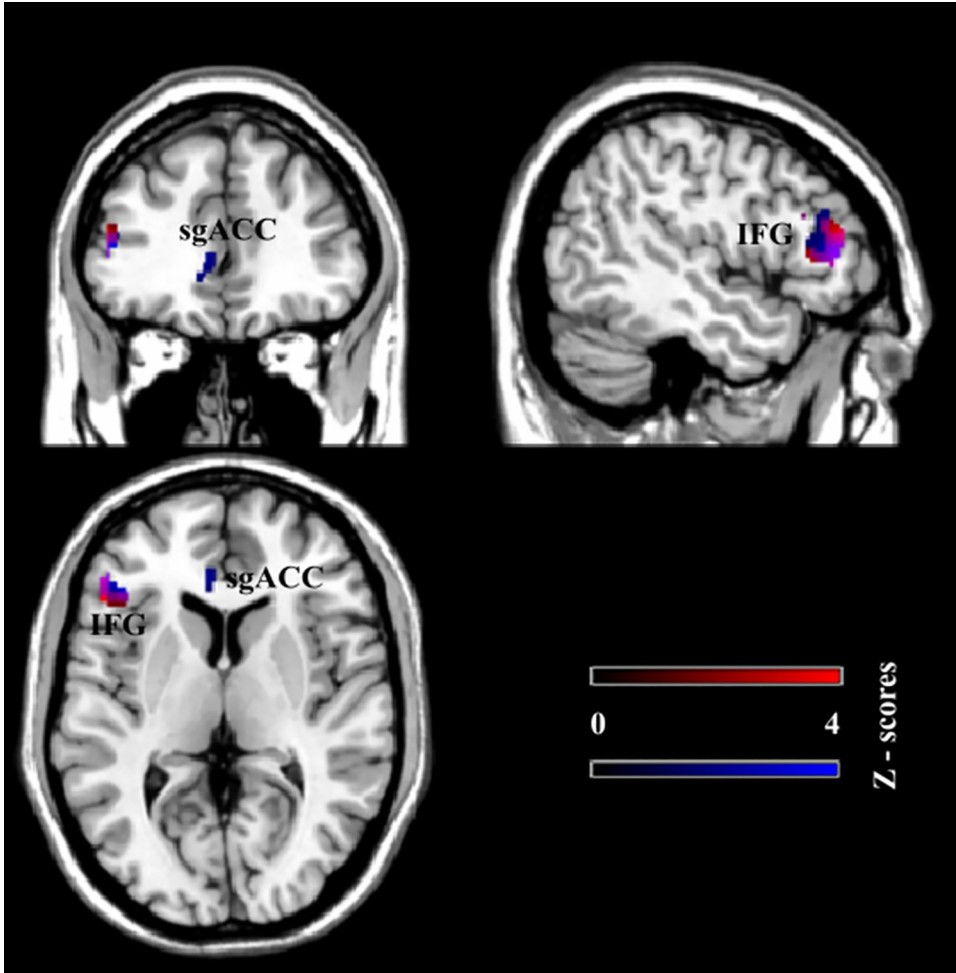

**Fig 3. Correlations of the CBF contrast of exclusion minus inclusion with subjective feelings of being rejected and victim scores positive correlations of the CBF contrast of exclusion minus inclusion with subjective feelings of being rejected (NTS, red) and victim scores (BVF-K, blue).** Overlap occurs in left IFG.

this brain area was activated. Another positive relationship between the number of victim experiences and differential activation during social exclusion was also observed in the pars opercularis of the left inferior frontal gyrus. Involvement of this region has repeatedly been reported to mediate inner speech [71] and may therefore reflect inner verbal monitoring and/ or coping with this situation as inner speech supports self-reflection and emotional regulation. For example, the thought like "Being rejected by others" during the exclusion session might enhance feelings of self-blame (e.g. "Why did they exclude me?" or "What did I do wrong?"). Morin [71,72] suggested that activity of left inferior frontal gyrus might reflect a variety of cognitive processes underlying inner speech. Similar to sgACC activation, the positive linear relationship may therefore reflect more pronounced mentalizing and/or rumination as a function of prior bullying experience. Compared with studies in adults, differential activation of the more dorsal aspects of anterior cingulate cortex [e.g. 41] did not survive the strict statistical thresholding as applied here. The lack of activation of this region might be due to an age-specific functional involvement of this area, in which emotion regulation in adolescents does not necessarily involve the dACC. This region has been associated with cognitive control [73], rather than emotional coping. Sebastian et al. [38] also showed age-related differences in

emotional regulation abilities, especially in social exclusion situations, which suggests that adolescents and adults might apply different strategies when coping with social exclusion: more "emotion-related" in adolescents and more "cognitive" in adults.

## Conclusion, limitations and future directions

The present study in teenagers with previous exposure to bullying in real life situations revealed activation in sgACC and other frontal structures in response to social exclusion similar to earlier work in adolescents without bullying experiences. Most importantly, it showed for the first time that the extent of bullying experience among teenagers was related to differential activation in sgACC and IFG, areas known to be active during the processing of feelings of social exclusion. This provides evidence that bullying experience increases sensitivity to signals of social exclusion.

When interpreting the results of our study, several limitations should be considered. In the present study, there was the lack of positive relationship between severity of depressive symptoms and frequency of prior experience of being a victim of bullying. This may be due to the rather low level of bullying experience and relatively low sum scores of the depression scale DIKJ in our sample. It would be interesting to see how the results in the present sample compares to a sample of more affected adolescents in a further replication study. Furthermore, future research including larger sample sizes are necessary to validate the current results at the neural level. Finally, psychological validity of the Cyberball paradigm might be somewhat reduced by repeating the same task condition three times, but averaging across task repetitions was advantageous for increasing signal-to-noise ratio of data analysis. Please note that ratings of feeling of belonging in the exclusion vs. inclusion conditions and post-experimental debriefing indicated that the paradigm induced the expected feelings of exclusion despite task repetitions.

The results of the present study could help to establish and evaluate school-based prevention programs. It could be investigated, for instance, whether increased sensitivity to social exclusion after bullying victim experience can be remediated through interventions. More future research is also needed to elucidate the mechanisms and mediating factors giving rise to mental disorders like depression after bullying victim experience.

## Acknowledgments

The authors thank Elke Scholz for recruiting of participants and her help.

## Author Contributions

**Conceptualization:** Markus Kiefer, Eun-Jin Sim, Rebecca Brown, Manfred Spitzer, Georg Grön.

**Data curation:** Markus Kiefer, Eun-Jin Sim, Sabrina Heil, Rebecca Brown, Manfred Spitzer, Georg Grön.

**Formal analysis:** Markus Kiefer, Eun-Jin Sim, Sabrina Heil, Bärbel Herrnberger, Georg Grön.

**Investigation:** Markus Kiefer, Eun-Jin Sim, Sabrina Heil, Georg Grön.

**Methodology:** Eun-Jin Sim, Sabrina Heil, Rebecca Brown, Bärbel Herrnberger, Georg Grön.

**Project administration:** Manfred Spitzer, Georg Grön.

**Resources:** Markus Kiefer, Eun-Jin Sim, Rebecca Brown, Bärbel Herrnberger, Manfred Spitzer, Georg Grön.

**Software:** Eun-Jin Sim, Rebecca Brown, Bärbel Herrnberger, Georg Grön.

**Supervision:** Markus Kiefer, Bärbel Herrnberger, Manfred Spitzer, Georg Grön.

**Validation:** Markus Kiefer, Eun-Jin Sim, Bärbel Herrnberger, Manfred Spitzer, Georg Grön.

**Visualization:** Markus Kiefer, Eun-Jin Sim, Georg Grön.

**Writing – original draft:** Markus Kiefer, Eun-Jin Sim, Sabrina Heil, Georg Grön.

**Writing – review & editing:** Markus Kiefer, Eun-Jin Sim, Bärbel Herrnberger, Georg Grön.

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
