## [Decision Letter · Decision Letter 0]

3 Mar 2021

PONE-D-21-02322

Neural signatures of bullying experience and social rejection in teenagers

PLOS ONE

Dear Dr. Kiefer,

Thank you for submitting your manuscript to PLOS ONE. After careful consideration, we feel that it has merit but does not fully meet PLOS ONE’s publication criteria as it currently stands. Therefore, we invite you to submit a revised version of the manuscript that addresses the points raised during the review process.

We look forward to receiving your revised manuscript.

Kind regards,

Haijiang Li

Academic Editor

PLOS ONE

Journal Requirements:

Reviewers' comments:

Reviewer's Responses to Questions

**Comments to the Author**

1. Is the manuscript technically sound, and do the data support the conclusions?

Reviewer #1: Yes

Reviewer #2: Partly

2. Has the statistical analysis been performed appropriately and rigorously? 

Reviewer #1: Yes

Reviewer #2: No

3. Have the authors made all data underlying the findings in their manuscript fully available?

Reviewer #1: Yes

Reviewer #2: No

4. Is the manuscript presented in an intelligible fashion and written in standard English?

Reviewer #1: Yes

Reviewer #2: Yes

5. Review Comments to the Author

Reviewer #1: The Introduction is generally well written. However, I advise the authors to do a more critical review of the literature and highlight what the limitations are in the relevant studies and what unique contribution this study would make.

Was the study approved by the IRB of the authors' affiliation? Describe more explicitly how the study met all ethical standards. Also, it would be good to have more information about the data collection procedures.

More information is needed on the instruments used in the study. For example, were they validated?

The analyses were appropriate for the study, it seems. However, I do not have the expertise in the analyses.

The Discussion section is generally well written; however, it would be helpful if the authors were explicit about the implications for future research and practice.

Reviewer #2: In one imaging study, the authors examined the impact of social exclusion during the Cyberball game on neural activity among schoolchildren aged 12 to 15 years. The authors also examined history of victimization from bullies as a potential moderating factor. I thought that there was much to like about this manuscript. The topic was interesting, the authors employed a powerful methodology for yielding neural data, the authors replicated some previous work, and the findings linking victimization to increased vulnerability were novel. I believe that this manuscript is appropriate for PLoS ONE. However, there are a few concerns that I think should be addressed prior to publication.

My primary concerns are methodological/statistical in nature. First, the authors could do more to add clarity to the research design to help the reader critically evaluate the authors’ findings. For instance, the sample size is small (although not unusually small given the nature and expense of the data collection method; Poldrack et al., 2017). However, fMRI research also tends to suffer from poor task-reliability (Elliott et al., 2020) which, when combined with small samples, can seriously undermine work in this area. Again, I understand that fMRI is an expensive method to employ; however, with that said, the authors should explicitly state and justify their decision rules for recruiting the sample size that they collected and address implications for statistical power by providing a power analysis for their sample size (Funder et al., 2014; Simmons et al. 2011). Furthermore, I encourage the authors to provide task-related reliabilities (e.g., test-retest intraclass correlations) to help readers critically evaluate the authors’ findings (Parsons et al., 2019). If the fMRI measures exhibit poor reliability, change scores (e.g., inclusion-exclusion contrasts) may not be the ideal analysis strategy (Cooper et al., 2019; Infantolino et al., 2018).

In terms of the statistical analysis of the data, I do appreciate the amount of detail that the authors provide when reporting their results. However, I have a number of suggestions that may aid in clarity and transparency. First, in addition to task-related reliabilities for the fMRI measures, general descriptives (Ms, SDs) and reliabilities (e.g., Cronbach’s alpha) would be helpful for each of the self-report measures (either in a table or in text). This would aid the reader in interpreting the distributional properties of those variables. In addition, it would be helpful if the authors could present some effect size metric (e.g., Cohen’s d) for the t-tests (lines 282-286). With all that said, the authors state that data for the study cannot be shared due to IRB restrictions. Why can’t the authors make deidentified data available?

During the experiment, participants engaged in 9 runs of the Cyberball game, with conditions being presented in a pseudo-random fashion. This seems like an advantageous presentation, experimentally, over fMRI studies that always present the exclusion trials last; however, I’m concerned about the believability and ecological validity of the task. For instance, after experiencing exclusion, future inclusion may appear unrealistic. Can the authors speak to this concern? Did participants express skepticism about their virtual partners?

Finally, the authors do not appear to present explicit, directional hypotheses. The authors predict that “neural responses to social exclusion (relative to social inclusion) were correlated with individually different levels of prior bullying experience”, but this hypothesis is rather vague (line 116). Did the authors expect a direction for the association? If so, state it. The introduction would also benefit from a stronger theoretical justification for the hypothesis that bullying experience will serve as a moderator. However, if the study was more exploratory in nature, that’s fine too but just make that explicit and transparent for the reader to avoid the impression of HARKing (Hypothesizing After Results are Known; Kerr, 1998).

One minor concern is that there were a few grammatical errors present in the manuscript (e.g., “For examples”; line 417). The manuscript would benefit from a thorough proofreading.

References:

Cooper, S. R., Jackson, J. J., Barch, D. M., & Braver, T. S. (2019). Neuroimaging of individual differences: A latent variable modeling perspective. Neuroscience & Biobehavioral Reviews, 98, 29–46.

Elliott, M. L., Knodt, A. R., Ireland, D., Morris, M. L., Poulton, R., Ramrakha, S., ... & Hariri, A. R. (2020). What is the test-retest reliability of common task-functional MRI measures? New empirical evidence and a meta-analysis. Psychological Science, 31, 792-806.

Funder, D. C., Levine, J. M., Mackie, D. M., Morf, C. C., Sansone, C., Vazire, S., & West, S. G. (2014). Improving the dependability of research in personality and social psychology: Recommendations for research and educational practice. Personality and Social Psychology Review, 18, 3-12.

Infantolino, Z. P., Luking, K. R., Sauder, C. L., Curtin, J. J., & Hajcak, G. (2018). Robust is not necessarily reliable: From within-subjects fMRI contrasts to between-subjects comparisons. NeuroImage, 173, 146–152.

Kerr, N. L. (1998). HARKing: Hypothesizing after the results are known. Personality and social psychology review, 2, 196-217.

Parsons, S., Kruijt, A.-W., & Fox, E. (2019). Psychological Science Needs a Standard Practice of Reporting the Reliability of Cognitive-Behavioral Measurements. Advances in Methods and Practices in Psychological Science, 2, 378–395.

Poldrack, R. A., Baker, C. I., Durnez, J., Gorgolewski, K. J., Matthews, P. M., Munafò, M. R., Nichols, T. E., Poline, J.-B., Vul, E., & Yarkoni, T. (2017). Scanning the horizon: towards transparent and reproducible neuroimaging research. Nature Reviews. Neuroscience, 18, 115–126.

Simmons, J. P., Nelson, L. D., & Simonsohn, U. (2011). False-positive psychology: Undisclosed flexibility in data collection and analysis allows presenting anything as significant. Psychological science, 22, 1359-1366.

6. PLOS authors have the option to publish the peer review history of their article (what does this mean?). If published, this will include your full peer review and any attached files.

Reviewer #1: No

Reviewer #2: No

---

## [Author Response · Author response to Decision Letter 0]

19 May 2021

Comments by Reviewer #1:

The Introduction is generally well written. However, I advise the authors to do a more critical review of the literature and highlight what the limitations are in the relevant studies and what unique contribution this study would make. 

Response: We thank this reviewer for her or his positive feedback and for drawing our attention to this lack of information. As suggested, we have re-written the introduction and the conclusion. We now better review relevant critical studies, describe unique aspects of our study and its limitations. This information is now given in the introduction (pp. 3-5) and in the conclusion, renamed “Conclusion, Limitations and Future Directions” (pp. 20-21) in the revised manuscript. The conclusion section was renamed because of restructuring of substantial parts in the revised manuscript.

Was the study approved by the IRB of the authors' affiliation? Describe more explicitly how the study met all ethical standards. Also, it would be good to have more information about the data collection procedures. 

Response: We apologize for not having provided sufficient detail with regard to the description of how we met the ethical standards. We obtained an approval from the local ethics committee of Ulm University Germany (https://www.uni-ulm.de/en/einrichtungen/ethikkommission-der-universitaet-ulm/) prior to our investigation. We additionally obtained the approval of a scientific study or a research project at public schools in the administrative district form the regional council of state education authority in Tübingen Germany (https://rp.baden-wuerttemberg.de/rpt/abt7/ref71/) because all participants in our study were schoolchildren aged 12 to 15 years, i.e. minor children We have re-written this information in the participants section (p. 6, line 138 ff) more clearly. Furthermore, we added more information of reasons why the data for the present study cannot be shared publicly in the Data Availability statement. We would like to note that the data in our study cannot be shared publicly because of the European General Data Protection Regulation (GDPR). In particular, when running the study, we did not obtain a formal written and signed consent from the caregivers of the adolescent participants, which regulates the transfer of biological material, including public sharing of brain imaging data. In order to clarify this issue, we have re-written the Data Availability statement: “Data cannot be shared publicly because of the European General Data Protection Regulation (GDPR), which does not allow transfer of biological material including brain imaging data without explicit consent of the participants. However, data are available from the corresponding author (contact via markus.kiefer@uni-ulm.de) or from the Data management assistant at the Section for Cognitive Electrophysiology within the Department of Psychiatry III in the University Ulm (contact via silvia.zischler@uni-ulm.de) for researchers who meet the criteria for access to confidential data by the Ethics Committee (contact via ethic-kommission@uni-ulm.de)”.

We also inserted more information about the data collection procedures in the materials and methods section (pp. 6 – 8). 

More information is needed on the instruments used in the study. For example, were they validated?

Response: We are thankful for these valuable suggestions. As suggested by this reviewer, we inserted further information on instruments in the questionnaires subsection (pp. 7 - 8 line 158 ff). 

The analyses were appropriate for the study, it seems. However, I do not have the expertise in the analyses. 

Response: We thank this reviewer for this positive feedback.

The Discussion section is generally well written; however, it would be helpful if the authors were explicit about the implications for future research and practice.

Response: Thank you for drawing our attention to these issues. As suggested, we now additionally discuss implication of the results and directions for future research in more detail. We correspondingly re-structured the conclusion, renamed “Conclusion, Limitations and Future Directions” (pp. 20-21) in the revised manuscript. 

Comments by Reviewer #2:

In one imaging study, the authors examined the impact of social exclusion during the Cyberball game on neural activity among schoolchildren aged 12 to 15 years. The authors also examined history of victimization from bullies as a potential moderating factor. I thought that there was much to like about this manuscript. The topic was interesting, the authors employed a powerful methodology for yielding neural data, the authors replicated some previous work, and the findings linking victimization to increased vulnerability were novel. I believe that this manuscript is appropriate for PLoS ONE. However, there are a few concerns that I think should be addressed prior to publication.

Response: We thank this reviewer for the positive evaluation.

My primary concerns are methodological/statistical in nature. First, the authors could do more to add clarity to the research design to help the reader critically evaluate the authors’ findings. For instance, the sample size is small (although not unusually small given the nature and expense of the data collection method; Poldrack et al., 2017). However, fMRI research also tends to suffer from poor task-reliability (Elliott et al., 2020) which, when combined with small samples, can seriously undermine work in this area. Again, I understand that fMRI is an expensive method to employ; however, with that said, the authors should explicitly state and justify their decision rules for recruiting the sample size that they collected and address implications for statistical power by providing a power analysis for their sample size (Funder et al., 2014; Simmons et al. 2011). 

Response: When planning the study at the beginning of the year 2017 we were not in the position to find reliable effect sizes from previous studies that would have permitted us to rely on for computing the necessary sample size to reach statistical power of at least .80. This was mainly due to the combination of several factors, that is, the novelty to engage MR-based perfusion imaging of the Cyberball-task in combination with schoolchildren who were also likely to have experienced some kind of bullying. Consequently, the rationale for setting the sample size was indeed driven by the article published by Poldrack et al. in January 2017 in Nature Neuroscience, where it was reported that nowadays (then) around 29 participants represent the median of usual sample sizes. We decided to include at least 30 schoolchildren, but had to learn soon that the interaction between children, teachers and particularly parents was not without complications, and did impose substantial hindrances which made it rather difficult for us to enclose the intended sample size within reasonable time. In the end 29 schoolchildren could be included, however, only because the time frame for acquisition was stretched several times. Unfortunately, during the analysis of this data it came out that three of them presented a history of neurological disorder, brain injury or neuroanatomical abnormalities. Two further participants were to exclude due to excessive head movements greater than 1.5 mm during MR imaging, resulting in a final sample of 24 participants. The rationale for determining the sample size in our study is now given on p. 6 lines 138- 144 in the revised manuscript.

Nevertheless, while the ideally a priori computation of sample sizes can no longer be done as the study is already closed we have spent some effort to compute a posteriori the power achieved with the present sample for all statistical comparisons we present in the manuscript. This certainly is not optimal but at least gives some hints that our study was not underpowered. In order to do so, we used the software package G-Power 3.1 (Faul, F., Erdfelder, E., Buchner, A., & Lang, A.-G. (2009). Statistical power analyses using G*Power 3.1: Tests for correlation and regression analyses. Behavior Research Methods, 41, 1149-1160) and computed Cohen’s d effect sizes from the smallest effect which we had observed throughout all our analyses we show in the manuscript. This value refers to the relationship between condition related perfusion changes (exclusion minus inclusion) and individual expressions of the BVK victim scale in the subgenual anterior cingulate (see Table 4 in the manuscript, last line). The estimated effect size was Cohens’s d of 0.54 which resulted in an estimated post-hoc power of 0.82 given the present final sample size of 24 participants and an a priori level of significance of p < 0.05. Since all other effect sizes were greater than that, it is not implausible to say the estimated post-hoc power was at least 0.82.

References: 

Faul, Franz & Erdfelder, Edgar & Buchner, Axel & Lang, Albert-Georg. (2009). Statistical Power Analyses Using G*Power 3.1: Tests for Correlation and Regression Analyses. Behavior research methods. 41. 1149-60. 10.3758/BRM.41.4.1149.

Poldrack, R. A., Baker, C. I., Durnez, J., Gorgolewski, K. J., Matthews, P. M., Munafò, M. R., Nichols, T. E., Poline, J.-B., Vul, E., & Yarkoni, T. (2017). Scanning the horizon: towards transparent and reproducible neuroimaging research. Nature Reviews. Neuroscience, 18, 115–126.

Furthermore, I encourage the authors to provide task-related reliabilities (e.g., test-retest intraclass correlations) to help readers critically evaluate the authors’ findings (Parsons et al., 2019). If the fMRI measures exhibit poor reliability, change scores (e.g., inclusion-exclusion contrasts) may not be the ideal analysis strategy (Cooper et al., 2019; Infantolino et al., 2018).

During the experiment, participants engaged in 9 runs of the Cyberball game, with conditions being presented in a pseudo-random fashion. This seems like an advantageous presentation, experimentally, over fMRI studies that always present the exclusion trials last; however, I’m concerned about the believability and ecological validity of the task. For instance, after experiencing exclusion, future inclusion may appear unrealistic. Can the authors speak to this concern? Did participants express skepticism about their virtual partners?

Response: We have taken the liberty to combine here both comments you have made in your review at different places, as they share the same context, that is, the issue of replication against the background of the novelty of engaging MR-based perfusion imaging to measure neural activation during the different Cyberball conditions. 

From our previous experiences with MR-based perfusion imaging (although 2D pseudo-continuous arterial spin-labeling (pCASL) instead of the newer 3D pCASL applied here), we knew that generally contrast-to-noise ratio is markedly less than in typical BOLD imaging. Nevertheless, we decided to use pCASL since it obviously has some advantages over BOLD imaging when it comes to measure blocks of any functional challenge that lasts longer than the typical 30 seconds in BOLD imaging. This has mainly to do with computation of difference images succinct in time during pCASL which overcomes problems of autocorrelation and particularly signal-drift inherent in BOLD imaging. While the latter aspect represents the rationale for having preferred pCASL instead of BOLD imaging, the costs were that the number of repetitions or the number of repeated measurements were not known then, which is, why three different conditions were repeated three times each, although we were aware that repetitions might counteract ecological validity as you also stated above. 

So, in order to answer your second comment first we can say, that after completion of the experimental sessions, we had asked all participants and they reported to have believed that they had played with two virtual partners throughout the entire experiment that is, none of the participants reported any scepticism with regard to the presence of their virtual partners. Furthermore, all participants felt to be included or excluded by their partners in the corresponding blocks. We have inserted this information in the results section (p. 13, lines 318-323). 

Next, we investigated how the three-times repetition of each condition might have confounded the overall signal change for the contrast exclusion minus inclusion given that the repetitions might have induced some kind of habituation, skepticism, or more generally reduced task engagement. Below, we have prepared a summary statistics for each of the eight result clusters that are reported in Table 2 of the manuscript representing the main result inferred from the exclusion minus inclusion contrast. For each cluster differential voxel intensities were averaged, and transformed into Cohen’s d effect sizes. The same procedure was also done for aggregated data, that is, averaging across all the three repetitions before conversion into Cohen’s d effect sizes as described above (see also Figure L1 in file "Response to Reviewers_PLOSONE_MK.docx").

Although not consistent, one can see that particularly the second repetition was detrimental to the inference of significant signal changes when compared against the first repetition. There are also almost parametric effects, that is, a linear decrease of effect sizes particularly in some of the larger clusters. This indicates that for these clusters the repetition of conditions was differently processed most likely in terms of habituation and that less repetitions would have been the better choice. Nevertheless, one can also see that for aggregated data effect sizes were greatest, compensate well for repetition effects and are in most instances greater than effect sizes for the first repetition supporting that the gain in contrast-to-noise from condition repetitions in MR-based perfusion imaging is greater than the loss introduced by psychological effects associated with repeating the same conditions. In other words: While present data indicate that psychologically validity may be compromised by repeating the same task condition, physically the benefit from averaging across repetitions is still advantageous. We discuss task repetition as possible limitation on p. 20 (lines 490-495) in the manuscript.

Nevertheless, the decrease in effect sizes, particularly for the second repetition also indicates that there is an issue with test-retest reliability due to two factors driving the variances in the signal of interest: psychologically, e.g. habituation, and physically insufficient measurements per repetition for computation of a more robust contrast-to-noise ratio. Consequently, intra-class correlation coefficients did never reach the height of .70 to indicate sufficient test-retest reliability. While this may appear unfortunate from a more psychometric perspective we would like to stress that the focus on this study was not on evaluating the test-retest reliability of different repetitions of the Cyberball conditions and that aggregating data across these repetitions did well compensate for the between-runs reliability issue. In the end, we have learned a lot from your valuable comment, which let us conclude that there is a complex and delicate trade-off between the numbers of physically necessary signal averages and associated psychological effects imposing the risk of habituation or even saturation. Thank you for drawing our attention to this issue.

In terms of the statistical analysis of the data, I do appreciate the amount of detail that the authors provide when reporting their results. However, I have a number of suggestions that may aid in clarity and transparency. First, in addition to task-related reliabilities for the fMRI measures, general descriptives (Ms, SDs) and reliabilities (e.g., Cronbach’s alpha) would be helpful for each of the self-report measures (either in a table or in text). This would aid the reader in interpreting the distributional properties of those variables. In addition, it would be helpful if the authors could present some effect size metric (e.g., Cohen’s d) for the t-tests (lines 282-286). 

With all that said, the authors’ state that data for the study cannot be shared due to IRB restrictions. Why can’t the authors make deidentified data available?

Response: Thanks for drawing our attention to this issue. As suggested, we now provide reliabilities (Cronbach’s alpha) of each questionnaire in the corresponding subsection (p. 7). We now also present descriptive statistics of the questionnaire data in a new table (Table 1). Accordingly, we rewrote the questionnaires subsection in the results section (p. 13). Cohen’s d for the t-tests is also provided now.

Concerning Data Availability and IRB restrictions, we apologize for not having provided sufficient detail in regards the description of the ethical standard procedures, and thank all reviewers for this remark. For social sciences and studies in related human subjects, we obviously have to obtain an approval from the local ethics committee of Ulm University Germany (https://www.uni-ulm.de/en/einrichtungen/ethikkommission-der-universitaet-ulm/) prior to our investigation. We additionally have to obtain the approval of a scientific study or a research project at public schools in the administrative district form the regional council of state education authority in Tübingen Germany (https://rp.baden-wuerttemberg.de/rpt/abt7/ref71/) because all participants in our study were schoolchildren aged 12 to 15 years, i.e. minor children, who also have bullying experience. We obtained all these approvals after the rigorous approval procedures. We have re-written the text in the questionnaires subsection (p. 6, line 138 ff). We would like to note that the data in our study cannot be shared publicly because of the European General Data Protection Regulation (GDPR). In particular, when running the study, we did not obtain a formal written and signed consent from the caregivers of the adolescent participants, which regulates the transfer of biological material, including public sharing of brain imaging data. In order to clarify this issue, we have re-written the Data Availability statement: “Data cannot be shared publicly because of the European General Data Protection Regulation (GDPR), which does not allow transfer of biological material including brain imaging data without explicit consent of the participants. However, data are available from the corresponding author (contact via markus.kiefer@uni-ulm.de) or from the Data management assistant at the Section for Cognitive Electrophysiology within the Department of Psychiatry III in the University Ulm (contact via silvia.zischler@uni-ulm.de) for researchers who meet the criteria for access to confidential data by the Ethics Committee (contact via ethic-kommission@uni-ulm.de)”.

Finally, the authors do not appear to present explicit, directional hypotheses. The authors predict that “neural responses to social exclusion (relative to social inclusion) were correlated with individually different levels of prior bullying experience”, but this hypothesis is rather vague (line 116). Did the authors expect a direction for the association? If so, state it. The introduction would also benefit from a stronger theoretical justification for the hypothesis that bullying experience will serve as a moderator. However, if the study was more exploratory in nature, that’s fine too but just make that explicit and transparent for the reader to avoid the impression of HARKing (Hypothesizing After Results are Known; Kerr, 1998).

Response: We thank the reviewer for this thoughtful suggestion. To our knowledge, our present investigation into experience of bullying in teenagers using MR measurement has not been carried out until now. Therefore, our study might be one of the first study regarding the neural correlate of social exclusion and bullying in teenagers. Given this exploratory character of our study with regard to the neural mechanisms related to bullying experience, we could not establish directional hypotheses for the association of bullying experience with neural responses during the Cyberball task. As the positive relation between bullying and depression is well established, we propose a positive correlation between bullying experience and depressive symptoms. As suggested, we rewrote the hypotheses section in the last paragraph of the introduction, in order to make the nature of our assumptions more explicit (p. 5, line 119-127). 

One minor concern is that there were a few grammatical errors present in the manuscript (e.g., “For examples”; line 417). The manuscript would benefit from a thorough proofreading.

Response: Thanks for drawing our attention to this mistake and others. We now have carefully checked English language and taken special care to avoid such mistakes in the revised manuscript.

---

## [Decision Letter · Decision Letter 1]

22 Jul 2021

Neural signatures of bullying experience and social rejection in teenagers

PONE-D-21-02322R1

Dear Dr. Kiefer,

We’re pleased to inform you that your manuscript has been judged scientifically suitable for publication and will be formally accepted for publication once it meets all outstanding technical requirements.

Kind regards,

Haijiang Li

Academic Editor

PLOS ONE

Additional Editor Comments (optional):

The revised anuscript has improved a lot for publication.

Reviewers' comments:

Reviewer's Responses to Questions

**Comments to the Author**

1. If the authors have adequately addressed your comments raised in a previous round of review and you feel that this manuscript is now acceptable for publication, you may indicate that here to bypass the “Comments to the Author” section, enter your conflict of interest statement in the “Confidential to Editor” section, and submit your "Accept" recommendation.

Reviewer #2: All comments have been addressed

2. Is the manuscript technically sound, and do the data support the conclusions?

Reviewer #2: Yes

3. Has the statistical analysis been performed appropriately and rigorously? 

Reviewer #2: Yes

4. Have the authors made all data underlying the findings in their manuscript fully available?

Reviewer #2: No

5. Is the manuscript presented in an intelligible fashion and written in standard English?

Reviewer #2: Yes

6. Review Comments to the Author

Reviewer #2: The authors have addressed my major concerns and have revised the manuscript with an eye towards transparency. Generally, I am satisfied with their responses, although I would like to confirm that their Cronbach alpha coefficients reported come from their own data. At the moment, p.7 of the manuscript reads like those values came from previous literature which doesn't tell us how those measures performed in their study. Otherwise, I don't have any other concerns.

7. PLOS authors have the option to publish the peer review history of their article (what does this mean?). If published, this will include your full peer review and any attached files.

Reviewer #2: **Yes: **Richard S. Pond, Jr.

---

## [Editor Report · Acceptance letter]

27 Jul 2021

PONE-D-21-02322R1 

Neural signatures of bullying experience and social rejection in teenagers 

Dear Dr. Kiefer:

I'm pleased to inform you that your manuscript has been deemed suitable for publication in PLOS ONE. Congratulations! Your manuscript is now with our production department. 

Kind regards, 

on behalf of

Dr. Haijiang Li 

Academic Editor

PLOS ONE